

# Harmonizing seismicity information in Central Asian countries: earthquake catalog and active faults

Valerio Poggi[1], Stefano Parolai[1,2], Natalya Silacheva[3], Anatoly Ischuk[4], Kanatbek Abdrakhmatov[5], Zainalobudin Kobuliev[6], Vakhitkhan Ismailov[7], Roman Ibragimov[7], Japar Karaev[8], Paola Ceresa[9], Paolo Bazzurro[9].

[1]National Institute of Oceanography and Applied Geophysics – OGS, Udine, Italy
[2]University of Trieste, Italy
[3]Institute of Seismology under MoES - IS, Republic of Kazakhstan
[4]IInstitute of Geology, Earthquake Engineering and Seismology, National Academy of Sciences of Tajikistan, Tajikistan
[5]Institute of Seismology of Kyrgyz Republic – ISNASKR, Kyrgyz Republic
[6]Institute of Water Problems, Hydropower Engineering and Ecology of the National Academy of Sciences of Tajikistan - IWPHE, Tajikistan
[7]Institute of Seismology of Uzbekistan – ISASUz, Uzbekistan
[8]UNDP Representative Office in Turkmenistan
[9]RED Risk Engineering + Development, Italy

*Correspondence to*: Valerio Poggi (vpoggi@ogs.it)

**Abstract.** Central Asian countries, which include Kazakhstan, the Kyrgyz Republic, Tajikistan, Turkmenistan, and Uzbekistan, are known to be highly exposed to natural hazards, particularly earthquakes, floods, and landslides. With the aim of enhancing financial resilience and risk-based investment planning to promote disaster and climate resilience in Central Asia, the European Union, in collaboration with the World Bank and the GFDRR, launched a regional program for "Strengthening Financial Resilience and Accelerating Risk Reduction in Central Asia" (SFRARR). Within this framework, a consortium of national and international scientific institutions was established and tasked with developing a regionally consistent multi-hazard and multi-asset probabilistic risk assessment. The overall goal was to improve scientific understanding on local perils and to provide local stakeholders and governments with up-to-date tools to support risk management strategies. However, the development of a comprehensive risk model can only be done on the base of an accurate hazard evaluation, the reliability of which depends significantly on the availability of local data and direct observations.

This paper describes the preparation of the input data sets required for the implementation of a probabilistic earthquake model for the Central Asian countries. In particular, it discusses the preparation of a new regional earthquake catalog harmonized between countries and homogenized in moment magnitude (Mw), as well as the preparation of a regional database of selected active faults with associated slip rate information to be used for the construction of the earthquake source model. The work was carried out in collaboration with experts from the local scientific community, whose contribution proved essential for the rational compilation of the two harmonized datasets.





# 1 Introduction

Except for the stable continental part of Kazakhstan, Central Asia is classified as a highly seismically active region. Large historical earthquake events have occurred, mostly caused by thrust and reverse-faults generated by the collision of the Eurasian and Indian plates (Ullah et al., 2015). Such compressional regime was responsible for the development of the Cenozoic belts of Tien Shan and Pamir, which accommodate a great part of the regional deformation (e.g., Abdrakhmatov et al., 1996; Zubovich et al., 2010) and where most of the seismicity occurs, often with earthquakes of magnitude larger than 7. Notable examples are the Verny (Ms = 7.3, 1887), Chilik (Ms = 8.3, 1889), Kemin (Ms = 8.2, 1911), Chatkal (Ms = 7.5, 1946) and Suusamyr (Ms = 7.3, 1992) earthquakes (Abdrakhmatov et al., 2003). The Kyrgyz Republic alone has been hit by 18 destructive earthquakes in the last 50 years, with up to 6.4 billion USD of potential economic losses estimated to be exceeded on residential buildings with a 10% probability in the next 50 years (Free et al., 2018). This seismically active region formally separates the more stable regions of the Tarim basin to the south and the Kazakh platform to the north, where a more moderate intraplate seismicity is observed but still capable of generating significant earthquakes.

On the territory of Turkmenistan, four seismically active regions can be identified: Turkmen-Khorasan, Balkhano-Caspian, Elbursky and Gaurdak-Kugitang. Strong destructive earthquakes took place, such as: Krasnovodsk catastrophic earthquake on July 8, 1895 (M=8.2); Germab earthquake on May 1, 1929 (M=7.2); Kazanjik earthquake on November 5, 1946 (M=7.0); Ashgabat catastrophic earthquake on the night of October 5-6, 1948 (M=7.3), Balkhan earthquake on December 06, 2000 G. (M=7.3). The larger seismicity is observed in the Turkmen-Khorasan and Balkhano-Caspian regions, with Ashgabat as the most seismically active area of the Turkmen-Khorasan region. Tajikistan is a seismically active region as well. Few destructive earthquakes are known, such as the Karatag earthquake in 1907 with MLH=7.4, the Sarez earthquake in 1911 with MLH=7.4, the Khain earthquake in 1949 with MLH=7.4, and the recent second Sarez earthquake in 2015 with Mw=7.2.

While most of the regional seismicity occurs within the first 40km of the crust, deep earthquakes have also been observed down to 300km depth in the Pamirs-Hindukush area (King et al., 1999). Although reverse and thrust source mechanisms predominate due to the local tectonic regime, strike-slip and -to a lower extent- normal mechanisms (or a combination thereof) are also present.

In this paper we describe the development of the input datasets required for the implementation of the earthquake component of a new probabilistic multi-risk model for the Central Asian countries of Kazakhstan, Kyrgyz Republic, Tajikistan, Turkmenistan, and Uzbekistan. This model is part of the EU-funded regional program "Strengthening Financial Resilience and Accelerating Risk Reduction in Central Asia", managed by the World Bank in collaboration with the Global Facility for Disaster Reduction and Recovery (GFDRR). In particular, this work focuses on i) the development of a new regional earthquake catalog, harmonized between countries and homogenized in moment magnitude (Mw), built using the most up-to-date information available from global and local sources and ii) the development of a selected dataset of major active lineaments that includes slip rate information, to complement the observed seismicity for the construction of a geodetically driven finite fault source model (see Poggi et al., 2023 for a comprehensive description of the probabilistic seismic hazard



model for Central Asia). It is important to note that the development of such regional datasets cannot occur without the contribution of experts from the local scientific community. Partnering with local government institutions and scientific

agencies is also an essential step to facilitate the consensus on models for possible integration into national seismic codes. Following this concept, the program consortium has partnered with the local scientific communities to share information and develop a review process that ranged from compiling the datasets to building the earthquake hazard model and discussing the respective results. In the following the creation process of the two main datasets is presented and discussed in detail.

## 2 A harmonized earthquake catalog for Central Asia

Nowadays, the compilation of a modern earthquake catalog with homogeneous magnitude information (e.g., Mw) is an essential step for the development of a probabilistic earthquake hazard model because it provides the basic information for evaluating the location, magnitude, and occurrence of potentially damaging future earthquakes.

The main notable examples of compilation and unification of earthquake catalogs in Central Asia were carried out within the framework of the international projects CASRI (from historical times until 2005) and EMCA (until 2009, Mikhailova et al.,

2015). Subsequently, the available information was supplemented with new data from SEME (Seismological Experimental and Methodical Expedition) and KNDC (Kazakhstan National Data Centre) for Kazakhstan and adjacent areas to support the development of a new national seismic zonation model and seismic microzonation of Almaty. However, a revision of the EMCA catalog (i.e., data prior to 2009) is needed. The epicenters of the earthquakes and the magnitude conversion relations used to create the catalog, including the description of intensity in moment magnitudes (Mw), need to be revised using the

latest information. Data after 2009 may be inconsistent in the catalogs of neighboring Central Asian countries due to differences in the development of observation networks and the use of different processing techniques.

In the following, we present the processing steps, key assumptions, and subjective decisions we made in creating a new harmonized earthquake catalog for Central Asia (hereafter HECCA) in moment magnitude (Mw) representation. The catalog was created by analyzing and combining publicly available global earthquake information (e.g., ISC-Reviewed, ISC-GEM,

GCMT, NEIC) with information from previous regional projects and local authorities of the states participating in the SFRARR project.

Although the catalog represents the best current snapshot of available earthquake information for the region, we nevertheless plan to make future additions to this compilation by gradually incorporating new data from local agencies, temporary networks, and regional projects as they become publicly available. For the compilation, we used a set of freely available and open-source

Python tools originally developed as part of the Global Earthquake Model Foundation to simplify and enable the process of future extensions (see https://github.com/klunk386/CatalogTool-Lite).



## 2.1 The harmonization approach

In order to create a homogeneous data set, it is usually necessary to collect and merge information from different sources. However, harmonizing data from different neighboring regions and homogenizing earthquake parameters (e.g., location

solutions, reported time, intensity scale) avoiding duplications is a rather complex process that requires establishing a set of objective criteria for selection, duplicate identification, merging, and conversion. This is often the case when different seismological agencies are reporting the same events but with different magnitudes (e.g., MI, Md, Ms). The same problem affects source location solutions when, for example, different networks use different earthquake phases, processing algorithms, or modeling assumptions (e.g., earth velocity structure).

In compiling the HECCA catalog, we proceeded in two steps. First, information from global sources and previous regional projects was collected, reviewed, and combined into a unique base compilation (the backbone), which was then supplemented by local/national datasets provided by consortium partners. It must be emphasized that the focus of this work is primarily on improving the catalog during the "instrumental period" (roughly after 1900, but especially after 1950, when modern analogue and then digital records became available). Rather, the historical events were imported directly from the EMCA compilation,

which is considered the authoritative source for this period, without further modification.

## 2.2 Input datasets

Authoritative global sources of information for creating the backbone part of the catalog include the ISC-GEM catalog, the ISC-Reviewed Bulletin, the Harvard-GCMT Bulletin, the USGS NEIC and the GEM Historical Catalog, and regional events from the EMCA catalog (**Table 1**). All datasets were preprocessed by filtering out events with a magnitude (any reported type)

below 2 and with an epicenter outside a buffer region of about 300 km around the five target states (**Figure 1**), since these events would not contribute significantly to the hazard. The national earthquake catalogs of the five local consortium partners (see **Table 3**) were then reviewed to supplement the backbone compilation.

| Source | N. of Events | Mag. Range | Mag. Type | Year Range | Depth Range |
| --- | --- | --- | --- | --- | --- |
| ISC-GEM | 1525 | 4.96 - 8.02 | Mw | 1906 - 2016 | 5.0 - 274.1 |
| ISV-Rev | 51093 | 2.0 - 8.4 | Various types | 1906 - 2018 | 0.0 - 441.4 |
| GCMT | 814 | 4.64 - 7.61 | Mw | 1976 - 2017 | 2.7 - 400.6 |
| USGS-NEIC | 15804 | 2.9 - 7.8 | Mw, Ms, mb | 1902 - 2020 | 0.0 - 400.57 |
| GEM-GEHC | 24 | 7.0 - 8.3 | Mw, Ms | 1052 - 1902 | 20.0 - 200.0 |
| EMCA – Hist. | 173 | 3.5 - 8.3 | Mlh | -2000 - 1898 | 3.0 - 180.0 |
| EMCA – Inst. | 30700 | 2.0 - 8.2 | Mlh | 1901 - 2009 | 0.0 - 404.0 |

**Table 1.** Summary of catalog sources used to create the HECCA backbone compilation (events selected within the buffer region surrounding the study area).

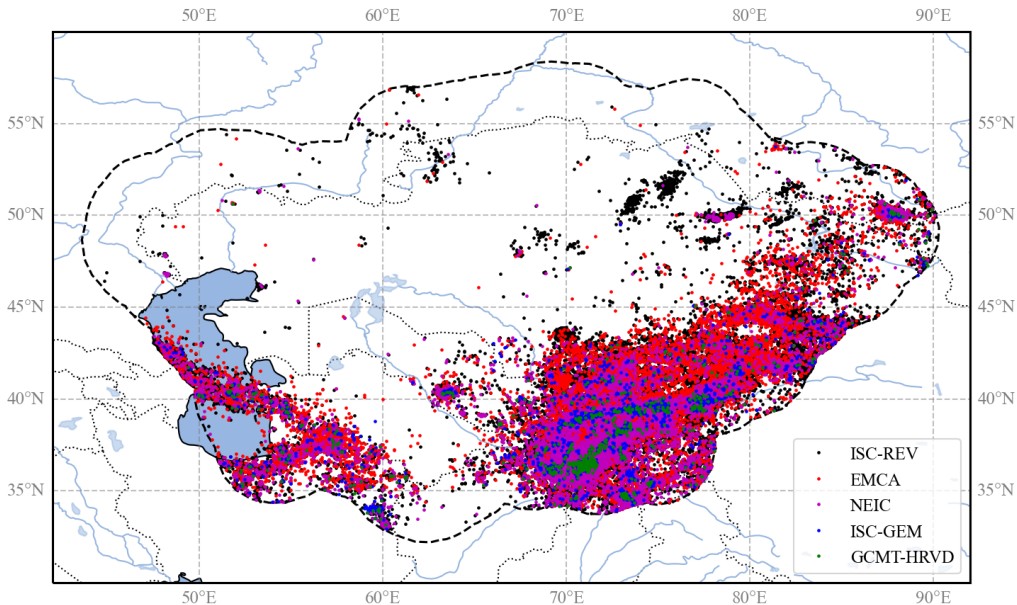

**Figure 1**. Distribution of epicenters of earthquake events from the main sources used to assemble the backbone compilation. The investigated area includes the five Central Asian countries, plus a buffer region of about 300 km around the country borders (black dashed line).

### 2.2.1 ISC-GEM

The ISC-GEM global instrumental catalog is an improved version of the bulletin of the International Seismological Centre (ISC, Storchak et al. 2013, 2015; Di Giacomo et al. 2018). The current version (version 7, published 2020-04-09) currently covers the period 1904-2016. The compilation benefits from an accurate relocation of earthquake events, performed using a single location procedure and a uniform velocity model (Bondar et al. 2015), while magnitudes have been all converted to the Mw scale according to the rules defined in Di Giacomo et al. (2015). At the global level, the catalog currently covers the magnitude range from about 5 to 9.5, with the magnitude record from 5.5 onwards being considered complete from 1935 onwards.

The ISC-GEM catalog is the primary and authoritative global source for the backbone catalog for Central Asia in the instrumental period. When selecting and merging events from different sources, the solutions for the ISC-GEM locality always have the highest priority over other solutions. In contrast, magnitude solutions have higher priority only when direct moment magnitude (Mw) estimates are not available (e.g., from the GCMT bulletin).

### 2.2.2 ISC reviewed bulletin

The reviewed version of the ISC bulletin (Storchak et al. 2017; www.isc.ac.uk) is used to add events not covered in the ISC-GEM catalog, especially for magnitudes below about 5.5, which are still relevant to earthquake hazard analysis.



The ISC reviewed bulletin contains multiple location and magnitude solutions (with different magnitude types) from different reporting agencies for each event. The Central Asia selection of the bulletin consists of 51093 events, with location solutions

from 33 agencies and magnitude solutions from 108 agencies (**Table 2**). ISC always provides a preferred ("prime") location solution, which is often - but not always - ISC's own solution. For catalog harmonization, we use the ISC prime location when available, which is derived from the same algorithm and velocity model used for the ISC-GEM catalog, while for defining magnitude we use a selection process based on agency prioritization rules, which are described in more detail in the next sections.


| Solution type | Agency (number of available solutions) |
|---|---|
| Location | ISC (41785), NNC (5646), BJI (552), IDC (478), KRNET (471), KNET (371), SOME (316), QUE (281), MOS (277), THE (241), EIDC (187), GUTE (109), NDI (77), THR (56), ASRS (53), IASBS (39), NEIC (30), ISS (26), CSEM (19), BCIS (17), DRS (15), CGS (8), OBM (6), PEK (6), MIRAS (6), MATSS (5), TIF (4), AZER (4), ISU (2), NEIS (2), MSSP (2), NORS (1), HFS1 (1) |
| Magnitude | IDC (92271), NNC (61850), ISC (25883), BJI (20887), NEIC (13595), MOS (13369), KRNET (9508), EIDC (4034), NEIS (2878), KNET (1376), NDI (1336), TEH (1282), QUE (1140), ASRS (1100), SOME (868), GCMT (845), CSEM (824), LDG (802), THR (762), USCGS (655), PEK (620), IASPEI (342), SZGRF (317), LAO (298), BGR (215), AZER (196), PAS (192), IASBS (116), EUROP (90), MIRAS (60), NAO (54), USGS;NEIC (51), HFS (51), ABE1 (44), GS (37), UPP (36), DRS (34), NORS (34), DSN (34), GUTE (34), OBM (31), STR (29), B&D (29), KIR (27), ZUR_RMT (27), P&S (25), BCIS (23), EVBIB (22), CGS (22), BRK (19), IPGP (18), BRK;NEIC (18), TEH;NEIC (17), COL (16), UPIES (15), ISN (14), DMN (13), MATSS (12), BRK;NEIS (12), KEW (11), MHI;NEIC (10), MAT (9), PAS;NEIC (9), KRAR (8), TIF (8), MSSP (8), UCDES (8), ROTHE (7), KISR (7), PAS;NEIS (7), NUR (6), HFS1 (6), PRA (6), AN2 (6), PSH;QUE (5), RSNC (5), MHI (4), USGS (4), OBN;NEIC (4), ZUR (4), PAL (4), SHL (3), ROM (3), LEDBW (3), STU (2), ISK (2), KLM (2), BJI;NEIC (2), GFZ (2), CNRM (2), LDSN (2), ABE3 (2), COP (2), TUL (1), KAR (1), IGS (1), CSE (1), BMO (1), PRE (1), PAL;NEIC (1), PDG (1), DNK (1), SFS (1), ISS (1), CSEM;NEIC (1), PMG (1), NDI;NEIC (1), CLL (1) |

**Table 2.** Location and magnitude solutions relative to each reporting seismological agency of the ISC-Reviewed bulletin in the study region.



For a comprehensive list and description of reporting agency codes and magnitude types refer to:

- http://www.isc.ac.uk/iscbulletin/agencies
- https://www.usgs.gov/natural-hazards/earthquake-hazards/science/magnitude-types

**2.2.3 GCMT bulletin**

The Global Centroid Moment Tensor catalog (GCMT, Ekström et al., 2012) is a collection of moment tensor solutions for

earthquakes with Mw > 4.5, from 1972 to 2013. While the solutions for the hypocenter come from external agencies (such as the ISC) and are therefore usually excluded from our analysis (or marked as duplicates), Mw solutions are always assumed to be authoritative reference estimates. The selection for Central Asia consists of 814 events with Mw between 4.6 and 7.6. Analysis of the moment tensor solutions for these events is also important to constrain the rupture mechanisms of the earthquake source model (see sections on defining rupture mechanisms).

**2.2.4 USGS – NEIC bulletin**

Although the International Seismological Centre bulletin is considered the definitive global archive of parametric earthquake data, the USGS National Earthquake Information Centre (NEIC) preliminary bulletin may provide useful additional information not yet reviewed by the ISC. The NEIC database generally has the lowest priority compared to previous compilations, both in terms of location and magnitude solutions.

**2.2.5 GEM historical earthquake catalog**

As in the case of the ISC-GEM catalog, the GEM historical earthquake catalog (GEM-GHEC, Albini et al. 2014) is an authoritative global source of information on historical earthquakes. The catalog covers events from about 1000 to 1903 and was compiled based on macroseismic intensity data and a review of the literature available worldwide (papers, reports, volumes). Unfortunately, GEM-GHEC has limited coverage of Central Asia, with only 24 events reported with magnitudes

greater than 7, most of which are recorded in the EMCA catalog.

**2.2.6 The EMCA catalog**

The EMCA (Earthquake Model of Central Asia) catalog (Mikhailova et al., 2015) contains information on 33620 earthquakes that occurred in the Central Asian countries (Kazakhstan, Kyrgyzstan, Tajikistan, Uzbekistan, and Turkmenistan) and represents the first major effort to harmonize catalog data in the region.

The EMCA catalog covers a period from 1000 to 2009 and is homogenized in surface wave magnitude Mlh for the horizontal component (Rautian et al. 2007). The Mlh magnitudes are not original estimates but were converted from either body wave





magnitude (mb), energy class (K), or Mpva (regional magnitude of body waves determined by the P-wave recorded by short-period instruments) using empirical regression analyses.

For the harmonization process, the catalog was divided into two main blocks, the pre-instrumental or historical (pre-1900) period and the instrumental (post-1900) period. Since the review of historical information is outside the scope of this project, all reported events prior to 1900 were considered authoritative sources for the creation of the new harmonized catalog. In contrast, location solutions from the instrumental period were thoroughly reviewed and, where necessary, replaced with solutions from the new catalog entries. Magnitude solutions were always considered authoritative over all other magnitude types (Ms, mb, Ml, Md), but not over Mw estimates from the moment tensor inversion and the ISC-GEM catalog.

### 2.2.7 Local earthquake datasets

The earthquake records from the backbone compilation were then integrated with information from the local earthquake catalogs provided by the national seismological agencies. These datasets are the result of regional earthquake monitoring conducted with temporary and national permanent seismic networks and are an essential complement to the information available worldwide, especially for the low magnitudes. The main characteristics of the national datasets reviewed for inclusion in the HECCA are listed in **Table 3**. It should be noted that several events of the local contributions were already available in the global sources and in the EMCA catalog. Therefore, the selection focused on identifying and including the missing events, especially for the most recent time interval, according to the harmonization procedures described in the following sections.

| Source | N. of Events | Mag. Range | Mag. Type | Year Range | Depth Range (km) |
|--------|-------------|------------|-----------|------------|------------------|
| Kazakhstan | 30930 | 2.1 - 8.3 (Ms) | Kp, Mlh, Ms | -250 - 2020 | 0 - 210 |
| Kyrgyzstan | 34434 | 2.2 - 7.7 (Ms) | Kr, Mlh, Ms | -250 - 2020 | 0 - 99 |
| Tajikistan | 66602 | 4.0 – 16.5 (Kr) | Kr | 1962-1991 | 0 - 350 |
| Uzbekistan | 1837 | 3.5 – 9.2 (Mlh) | Kr, Mlh | 1955-2020 | 0 - 35 |
| Turkmenistan | 7416 | 8.6 – 14 (Kp) | Kp, Mpv | 1997-2014 | 0 - 63 |

**Table 3.** Summary of local national sources used to supplement the final HECCA catalog (magnitude range refers to final conversion to Mw).

### 2.3 Duplicate finding

To create a unique catalog compilation, the first step is to identify the same events from the different input sources and merge them using a duplicate detection algorithm. Our approach is based on spatial and temporal matching of the reported hypocentral solutions within predefined windows, the length of which is tuned to the expected accuracy of the solution in each time range. For the current study, we determined an optimal time range of 15 seconds and a spatial distance of 60 km between solutions (**Figure 2**). This combination allowed us to capture over 95% of the duplicate events in the instrumental period (after 1900). Because this is an automated process, errors in identification are still possible. Because there is no unique window length that





allows all duplicate events in all catalogs to be captured without erroneously including a subset of independent events, an
additional magnitude range match condition was added to reduce the likelihood of false identifications. A condition of 1
magnitude unit difference was introduced as the maximum allowed gap between duplicate events.

Due to the limited extent of historical records (from EMCA and GEM-GEHC), the merging of historical data sources was done
manually.

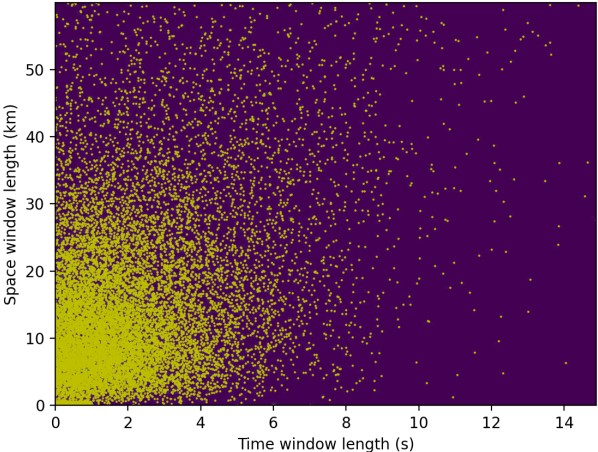

**Figure 2.** Temporal and spatial distance of events identified as duplicates between the ISC bulletin and the EMCA catalog. More than
95% of the events are covered by a 15-second, 60-km window, although the majority of events are within a 5-second, 25-km difference.

Once duplicate events are identified between the catalogs, the solutions are merged into a single event with multiple locations.
As a final step, the preferred location solutions are then selected according to ad-hoc priority rules (see **Table 4** for the main
backbone catalog contributions to location solutions, sorted by priority). It is worth noting that EMCA has a lower priority
compared to other reporting sources only for the location solutions. Indeed, a significant portion of EMCA events have low
spatial resolution (resulting in a "gridded" pattern in the distribution of epicenters). As mentioned earlier, reporting sources
such as ISC-GEM (and more recently ISC-rev) now provide reprocessed solutions that use newer and better performing
algorithms and regionally consistent velocity models.

| Source | ISC-GEM | ISC-Rev (prime) | GCMT | USGS-NEIC | EMCA |
|---|---|---|---|---|---|
| **Initial** | 1526 | 51093 | 814 | 15804 | 30700 |
| **Selection** | 1526 | 49751 | 0 | 1554 | 16156 |

**Table 4.** Number of events selected as preferred location solutions from the various input datasets used to create the backbone catalog.
Sources are sorted from highest (left) to lowest (right) priority rule.



## 2.4 Magnitude homogenization

A key point in the harmonization process is the representation of all available earthquake events with a uniform target magnitude. In this study, we use moment magnitude Mw (Hanks and Kanamori 1979) as the reference type because it is directly related to earthquake size and energy and there is no saturation at high magnitudes. However, events with a native

estimate of Mw (e.g., obtained directly from data) are limited (e.g., post-1976 for the GMCT catalog), so conversion from other scales is often required.

### 2.4.1 Agency selection

For magnitude homogenization we applied a magnitude agency selection criterion analogous to that used for the selection of the preferred location. In a first step, we examined the availability of different magnitude types from each available agency.

Subsequently, the most reliable agencies were selected and sorted according to specific priority rules. Prioritization was based on magnitude type (from higher to lower priority: Mw → Mlh → Ms → mpv → mb → Ml) and agency-specific selection criteria. **Table 5** provides the final list of magnitude types and agency priorities. Using these rules, a single magnitude estimate is then assigned to each event (**Table 6**).


| Group | Type | Agency |
|-------|------|--------|
| Mw | Mw* (all variants) | GCMT-NDK, GCMT, HRVD, HRVD-NEIC, NEIC, USGS, USGS-NEIC, MOS, ZUR_RMT, ISC-GEM |
| Mlh | Mlh | EMCA |
| Ms | MS, Ms, MSZ, Msz, Ms1 | ISC, IDC, MOS, BJI, SOME, NEIC, EIDC, NEIS, PEK, PAS |
| mpv | Mpv | NNC |
| mb | mb, mb1, Mb | ISC, IDC, MOS, NNC, KRNET, NEIC, NEIS, USGS, BJI, QUE, EIDC, USCGS |
| ml | ML, Ml, mL | IDC, EIDC, BJI, CSEM, TEH, THR |
| others | Md and unknown types | Not represented in the final compilation |

**Table 5.** Magnitude priority rules applied to the HECCA backbone catalog. Magnitude types, variants and reporting agencies are sorted from highest to lowest priority.






| Agency | N. of Events | Magnitude (relative occurrence) |
|---|---|---|
| EMCA | 29334 | Mlh (29334) |
| NNC | 23679 | mpv (23575) mb (104) |
| IDC | 4194 | MS (3516) mb (596) mb1 (54) ML (28) |
| ISC | 3855 | mb (2732) MS (1123) |
| USGS | 1407 | mb (1353) Mww (36) Mwr (18) |
| ISC-GEM | 1059 | Mw (1059) |
| KRNET | 906 | mb (906) |
| GCMT-NDK | 816 | MW (816) |
| BJI | 751 | ML (299) mL (244) Ms (147) mb (39) MS (22) |
| QUE | 360 | mb (360) |
| NEIS | 327 | mb (293) MSZ (21) MS (13) |
| NEIC | 302 | mb (239) Mwr (43) MS (10) MW (3) MSZ (3) Mww (3) Mw (1) |
| TEH | 254 | ML (254) |
| MOS | 246 | mb (131) MS (43) Mb (38) Ms (34) |
| CSEM | 231 | ML (231) |
| EIDC | 204 | mb (141) MS (62) mL (1) |
| SOME | 127 | MS (127) |
| USCGS | 54 | mb (54) |
| THR | 47 | ML (47) |
| GCMT | 45 | MW (45) |
| PEK | 43 | MS (43) |
| PAS | 37 | MS (37) |
| ZUR_RMT | 18 | Mw (18) |

**Table 6.** Number of events selected as preferred magnitude solutions from the different reporting agencies for the instrumental period (after 1900). Agencies are ordered by relative frequency of the events (from highest to lowest).

### 2.4.2 Magnitude conversion

As a final step in the construction of the catalog, all events with different magnitude types must be converted to a reference scale, in this case the moment magnitude Mw. For the conversion, we prefer to use robust, well-tested, and globally calibrated magnitude conversion relations for the most common magnitude scales (Ms, mb, Ml), whereas for the conversion of specific scales (Mpv and Mlh) to Mw, ad hoc relations were developed using an orthogonal regression approach (e.g., **Figure 3**). In



these models, the saturation limits of each scale were included as an additional physical constraint on the regression model to
stabilise the regression result. See **Table 7** for the complete list of conversion rules.

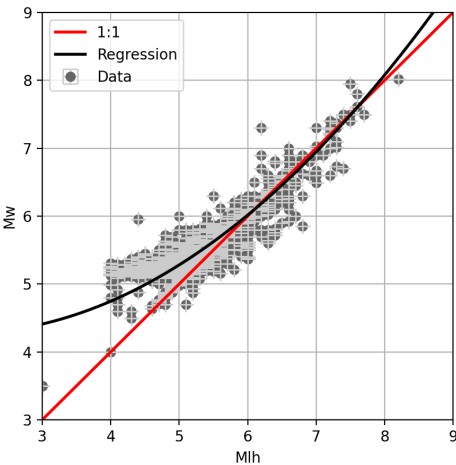 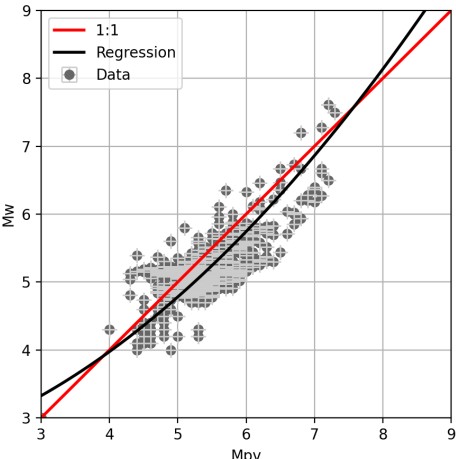

**Figure 3.** The magnitude conversion relationships developed for Mlh and Mpv scale to Mw by fitting $2^{nd}$ degree polynomial to observed
magnitude pairs using the orthogonal least squares regression technique (**Table 7**).

| Type | Conversion Rule |
|---|---|
| Mw | 1:1 |
| Mlh | $4.594 - 0.359M + 0.099M^2$ (this study) |
| Ms | Di Giacomo et al. (2015) – Exponential |
| Mpv | $2.311 + 0.104M + 0.078 M^2$ (this study) |
| mb | Weatherill et al. (2016) – Linear (NEIC calibration) |
| ml | Edwards et al. (2010) - Polynomial |
| Md and others unknown types | 1:1 |
| Kr (energy magnitude) | Bindi et al. (2011) |

**Table 7.** Magnitude conversion relations used for the homogenization of the HECCA catalog in Mw.

**2.5 Integration of local data**

The harmonization process (duplicate identification, location selection, magnitude conversion) was first performed on the
global and regional datasets to produce the backbone part of the harmonized catalog. The inclusion of local (national) datasets
in the backbone compilation was then done using the same integration criteria, but in a separate phase. Merging of the different
national contributions was done for each country individually, so that each dataset was assumed to be authoritative for its





territory and no additional priority rules were needed for selection. In addition, uniform rules for magnitude conversion were used, as indicated in **Table 7**.

## 2.6 The final compilation

The harmonized backbone catalog for Central Asia presently consists of 77376 events through 2020 and in the range 3.0<Mw<8.5 (see, e.g., **Figure 4**, **Figure 5** and **Figure 6**), with a minimum regional completeness of about Mw 4 to 4.5. Of

the total number of compiled events, about 10646 are from newly recorded local data (roughly 13% of the total). The historical period (pre-1900) is mostly covered by the EMCA catalog, while the instrumental period has been thoroughly revised and expanded by including new homogenous location solutions from global datasets, additional magnitude conversion relations and more recent events (e.g., after 2009) from regional datasets.

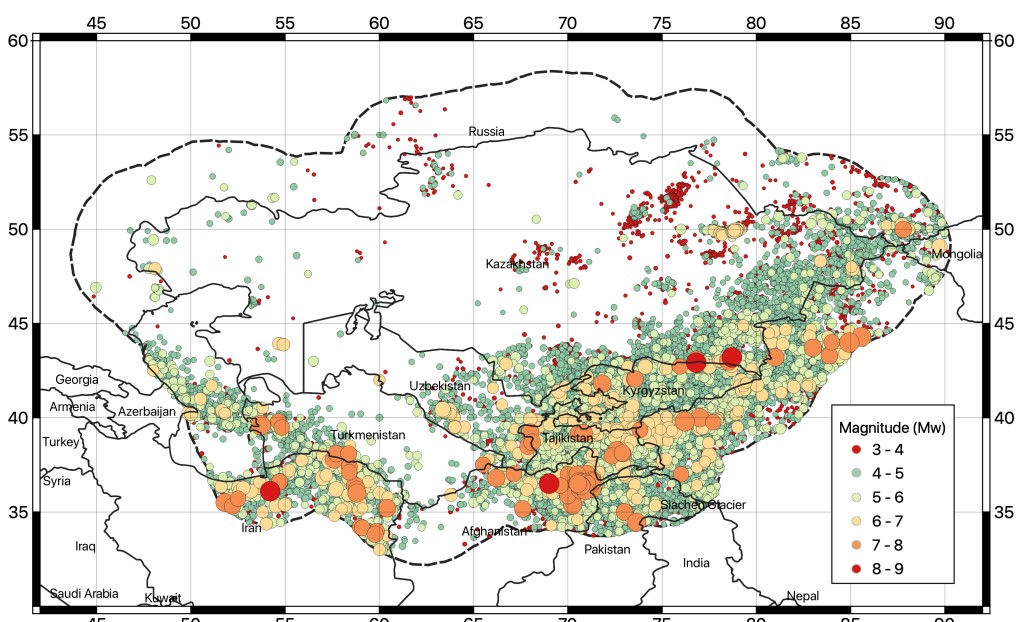

**Figure 4.** Geographic distribution of earthquake hypocenters (Mw>3) of the newly developed Mw harmonized catalog for Central Asia (HECCA).


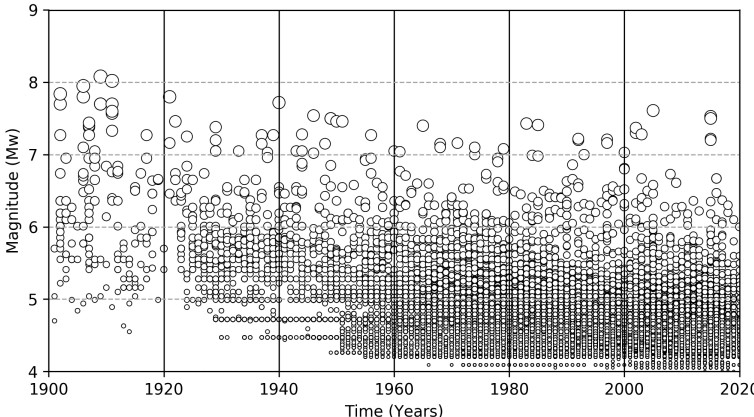

**Figure 5.** Time-magnitude distribution of the earthquake events of the HECCA catalog in the instrumental period (after 1900).

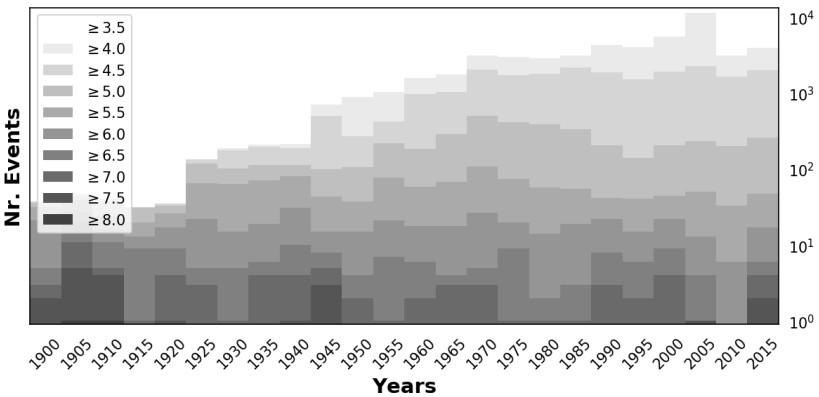

**Figure 6.** Number of events of the Central Asia catalog calculated for five-year windows in the period 1900-2015. Shading refers to bins with increasing magnitude threshold (cumulative).


### 2.7 Declustering

Probabilistic seismic hazard analysis assumes that earthquake events are independent and that their probability distribution corresponds to a Poisson process. In reality, earthquake catalogs are characterized by a proportion of correlated events that are highly interdependent in space and time. The clustering of correlated events may be of natural origin (e.g., the aftershocks
following a major event), triggered by anthropogenic activities in the natural environment (e.g., geothermal exploitation - extraction of thermal energy by pumping fluids from a geothermal reservoir and carbon sequestration - process of capturing and storing atmospheric carbon dioxide in an already depleted reservoir), or purely artificial (e.g., blasting, mining explosions). In all cases, these events must be removed so that the earthquake record is equivalent to a Poisson process. Declustering



techniques are usually used for this purpose. What remains can be considered as a collection of independent mainshocks (i.e.,
events with the largest magnitude in a cluster) of purely tectonic origin.

### 2.7.1 Natural events

In this study, aftershocks, foreshocks, and triggered events in all clusters are removed using a direct search approach, where
all events that are within a magnitude-dependent time window from the assumed mainshock (largest event in the cluster) are
considered dependent and then removed from the catalog. Several time-distance windows have been proposed in the literature.
We tested the algorithms of Gardner and Knopoff (1974), Uhrhammer (1986), and Grunthal (1985), each of which provided
different estimates of the relative aftershock fraction. By directly testing the performance of the three algorithms on the
HECCA (e.g., **Figure 7**, **Table 8**), both in terms of the geographic distribution of residual events and the variation in frequency
of occurrence, we selected Gardner and Knopoff (1974) as the approach that provides the most reasonable and balanced result
for Central Asia, as it is not too aggressive while being able to capture most of the dependent events.


| | All events | 3<Mw<4 | 4<Mw<5 | 5<Mw<6 | 6<Mw<7 | 7<Mw<8 |
|---|---|---|---|---|---|---|
| Before declustering | 77376 | 25178 | 47599 | 4060 | 444 | 91 |
| GardnerKnopoff | 24373 | 7398 | 14878 | 1774 | 248 | 71 |
| Uhrhammer | 49018 | 17191 | 29146 | 2337 | 272 | 68 |
| Grunthal | 14283 | 3654 | 8788 | 1539 | 228 | 70 |

**Table 8.** Number of earthquakes per magnitude bin from the non-declustered d catalog and using different declustering algorithms.

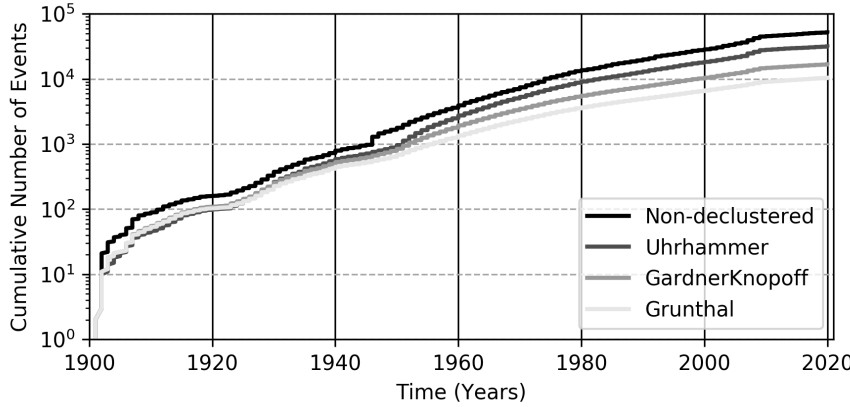

**Figure 7.** Cumulative number of events over time for the full (non-declustered) HECCA catalog and for the three catalogs obtained using
the three considered declustering algorithms.



### 2.7.2 Induced and artificial events

In principle, induced and artificial events caused by humans should be known from the beginning and could therefore be manually removed from the earthquake record. However, in the case of Central Asia, the record of these events is fragmented and often incomplete. Therefore, an alternative (and possibly automated) removal strategy needs to be introduced and applied. The main problem is that these events may overlap in time and space with existing background seismicity, which should not be modified to avoid biased estimation of local hazard.

Here, we applied a modification of the declustering algorithm used to clean up natural aftershocks, assuming that such artificial events are also highly clustered in space and time and that, at the same time, the largest events in the cluster are likely to be of natural origin. Based on a Gardner and Knopoff (1974) window, a variable scaling factor is then applied to the spatial and temporal extent of the window until an optimal tradeoff between cleaned events and remaining seismicity (compatible with the regional background) is found. After several trials, we determined the best scaling factor for the region to be 100. To avoid altering the earthquake record in areas not affected by man-made events, the procedure is applied only to buffer regions (polygons) with known anthropogenic activity. Currently, seven polygons have been identified and reported by the local partners of the consortium, five of which are located in the inner stable cratonic part of Kazakhstan and one in the more active region near the border with Kyrgyzstan (e.g., **Figure 8**). According to the proposed procedure, 558 events were identified as anthropogenic, which is about 2% of the original declustered catalog.

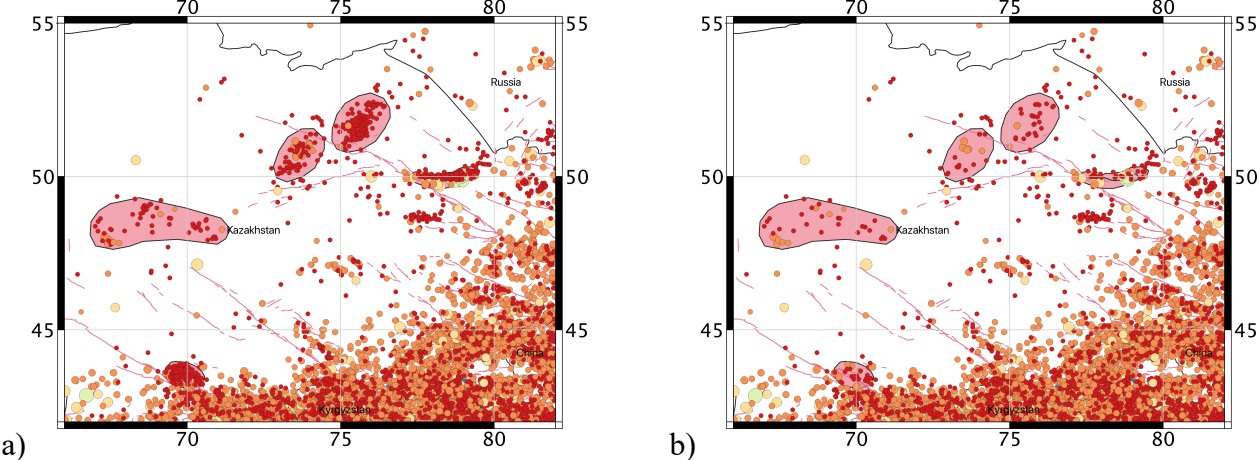

**Figure 8.** Example of application of the procedure to remove artificial events from the catalog. In pink, the polygons delineate areas of known anthropogenic activity.



## 3 An active fault dataset for finite source modeling

The inclusion of finite fault source models in probabilistic seismic hazard assessment is now becoming standard practice, as it
provides a convenient approach to better represent near-field ground motions when targeting specific and well-defined active
structures, thus complementing some of the limitations of distributed seismicity models. However, accurate modelling of
potentially seismogenic faults is only possible if sufficient information (fault geometry, kinematic parameters, displacement
rates) is available with sufficient confidence for the area under study (e.g., clear surface expression, known segmentation, well-
documented evidence of Quaternary activity or direct seismicity, etc.), which is not the case for most observed tectonic
lineaments. This section presents the construction of a dataset of active faults from existing regional compilations to be used
for the probabilistic seismic hazard assessment of Central Asia (Poggi et al., 2023).

### 3.1 The modelling strategy

The fault parametrization adopted in this study is determined by the requirements of the chosen source modelling strategy. We
use the modelling formalism of the OpenQuake engine (Pagani et al., 2014), an open-source seismic hazard and risk calculation
software developed, maintained, and distributed by the Global Earthquake Model (GEM) Foundation. However, finite fault
sources can be modelled in OpenQuake in different ways, depending on how accurate the fault representation should be. In
this study, we use the "simple fault" modelling approach (see "OpenQuake technical manual" for more details on modelling),
in which the three-dimensional fault geometry is approximated by extending the fault trace from the Earth's surface to the
lower seismogenic depth with an inclination equal to the dip angle (**Figure 9**). The complete list of modelling parameters
required for simple fault and the corresponding values used as reference in this work are summarized in **Table 9**.

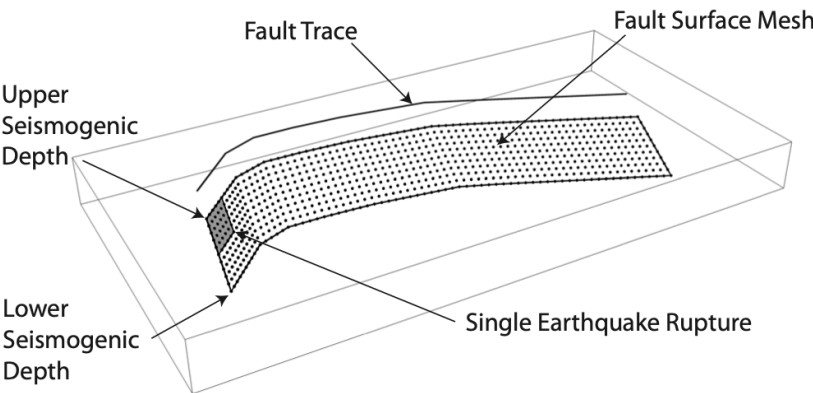

**Figure 9.** Simple Fault source in the OpenQuake engine (modified from "the OpenQuake-engine book: underlying hazard science").





| Parameter | Value |
|---|---|
| Fault trace | Taken from fault database (in geojson format) |
| Upper seismogenic depth (USD) | 0 (surface rupture) |
| Lower seismogenic depth (LSD) | Defined by applying Leonard (2014), with the additional constraint of not exceeding the maximum seismogenic depth of the source group |
| Dip angle | Extrapolated from geometry description of the fault database, following the Aki and Richards (1980) convention |
| Rake angle | Extrapolated from geometry description of the fault database, following the Aki and Richards (1980) convention |
| Magnitude frequency distribution (MFD) | Double-truncated Gutenberg-Richter (GR) distribution, with lower-bound magnitude fixed to M6.0 and upper-bound magnitude defined by applying Leonard (2014), with the additional constraint of not exceeding the maximum magnitude of the source group |
| Magnitude-area scaling relationship | Leonard (2014) |
| Rupture aspect ratio (length/width) | Fixed to 2.0 |
| Aseismic coefficient | Fixed to 0.1 |

**Table 9.** Summary of the essential parameters and the corresponding values used for the definition of a fault source model in Central Asia.

**3.2 Regional active fault datasets**

At the regional level, the most significant existing compilations that are uniform and consistent across Central Asian countries are the GEM Global Active Fault Database (GEM GAF-DB, Styron and Pagani, 2020, **Figure 10**) and the Active Fault Database of Eurasia (hereafter AFEAD, Bachmanov et al., 2017, Zelenin et al. 2022, **Figure 11**), which review and summarize most of the available information from published scientific studies for the target area.

In particular, the AFEAD database currently contains more than 20 thousand lineaments (faults, fault zones, and associated
structural shapes) that show evidence of recent displacement during the late Pleistocene and Holocene. For each mapped fault, the database reports morphological and kinematic information with quality indicators (four reliability classes from A to D, from most reliable to least reliable) and, where possible, an assessment of displacement rates (three ranks of late Quaternary movements). Conversely, only a limited number of faults from the GEM GAF database are sufficiently complete to be used for building fault source models (e.g., because of a lack of estimates of displacement rates). In direct comparison, these faults
are also included in the AFEAD database, so most of the information is shared between sources. For this reason, although the AFEAD database has some local inconsistencies that require some attention (e.g., in the segmentation of faults), at present time it is the primary information base for building the finite fault source model for this study.

The AFEAD database was then exported to an open format compatible with GEM GAF (in geojson format, see following section) to facilitate the comparison and the integration of additional information that may be derived from new local studies.

Such a compilation will be made openly available to encourage further development of the area.

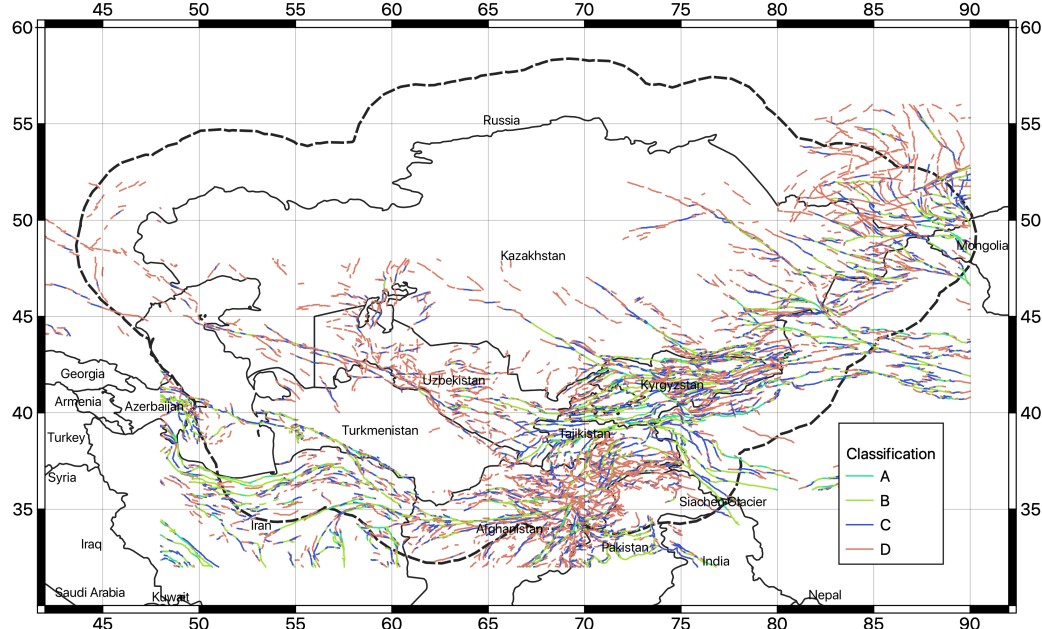

**Figure 10.** Traces of faults available in the database of Active Fault for Eurasia and adjacent regions (AFEAD).

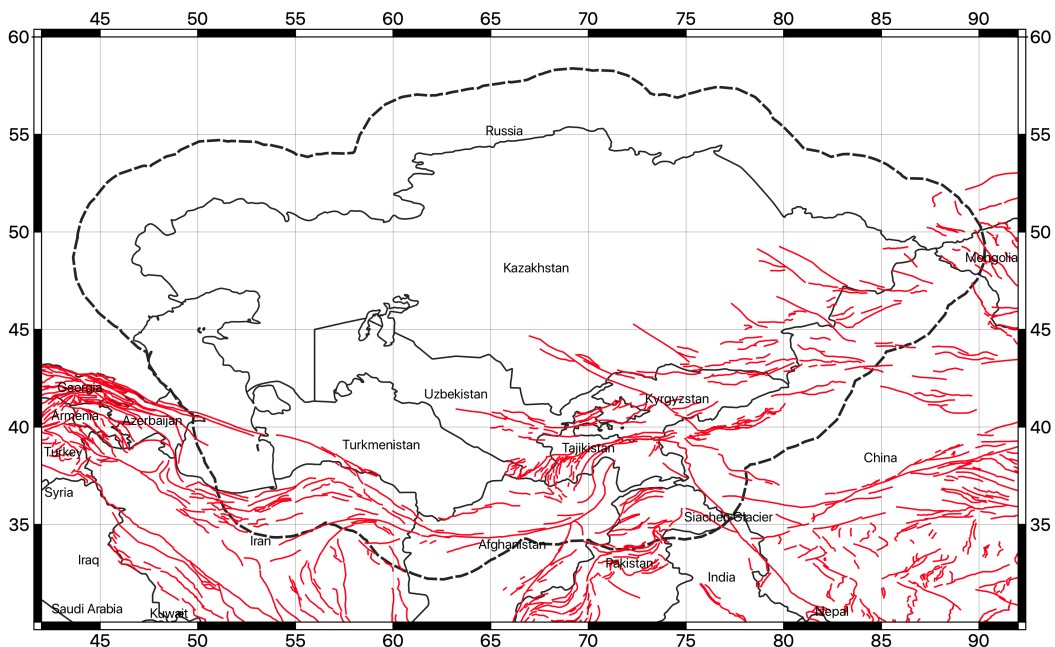

**Figure 11.** Traces of faults available in the global active fault database of GEM (GEM GAF-DB).

## 3.3 Format conversion and fault selection

To create the fault source model, the AFEAD database was first converted to an intermediate format compatible with the GEM
Global Active Fault Database. Such a format is basically required to build the OpenQuake source model using the Model
Building Toolkit from GEM, as was done in Poggi et al. (2023) for the development of a probabilistic earthquake hazard model
for Central Asia. Because it is in plain-text geojson format, it also has the added advantage of being easily maintained and
extended using common version control tools (e.g., Git) and GIS software (e.g., QGIS). However, translating the original
AFEAD database into the GEM format required a certain amount of interpretation, as not all parameters could be directly
assigned. In addition, only a subset of the parameters from GEM are used (see https://github.com/GEMScienceTools/gem-global-active-faults for a description of the GEM GAF format ).

The parameter conversion rules are described in **Table 10**. Note that any parameters not explicitly specified in the conversion
table were discarded during compilation. In addition, faults with missing required parameterization (e.g., unknown value for
the parameter SIDE) were not considered and therefore are not currently converted to the source model.

The most sensitive parameter of the conversion process is the net slip rate. The AFEAD database provides an approximate and
quite wide range of slip rates for each RATE class (1,2,3), which we converted to numerical values (in cm/y) by comparing
them to the slip rates reported in the GEM GAF database and from scientific literature. However, to account for the unavoidable
uncertainties associated with the conversion, three alternative rate conversion models were implemented, including a middle
estimate, an upper bound, and a lower bound, with the goal of using them for hazard calculation in a logic tree structure.



Only faults with reliability class A and B (independent evidence of activity in the form of kinematics and clear evidence of strong earthquakes) were explicitly considered, while classes C and D were discarded because of their unclear, incomplete, or inaccurate interpretation. This conservative choice could be relaxed in future analyses as additional information becomes available for Class C and D lineaments. The selected subset consists in 1444 individual fault segments, covering most of the active shallow crust currently affected by seismicity.


| GEM parameter | AFEAD parameter | Conversion convention |
|---|---|---|
| name | NAME | Same |
| slip_type | SENS1 | D=Dextral, S=Sinistral, T=Thrust, R=Reverse, N=Normal |
| average_dip | SENS1 | D=90°, S=90°, T=30° R=40°, N=60° |
| average_rake | SENS1 | D=180°, S=0°, T=90° R=90°, N=-90° |
| dip_dir | SIDE | Same |
| net_slip_rate | RATE | 3= (0.05, 0.1, 0.2), 2=(0.25, 0.5, 1.0), 1=(0.5, 1.0, 2.0) Values are (min, mean, max) slip rates in cm/y |
| reference | AUTH | Same |
| notes | TEXT | Same |
| -- | CONF | Only quality class A and B have been considered |

**Table 10.** Parameter conversion rules used to migrate the AFEAD database into the GEM GAF format.

**4 Conclusions**

This paper presents the creation of the datasets required for the calculation of a new probabilistic seismic hazard model for Central Asia under the SFARRR Regional Program ("Strengthening Financial Resilience and Accelerating Risk Reduction in Central Asia"). The main objective was to create a preparatory collection of data based on the most complete and up-to-date information available for the territory, consistent in its methodological construction and uniform for all Central Asian countries. The homogenization of input datasets was definitely one of the most critical and challenging steps in the analysis, as the

quality, completeness, and reliability of the data required to build the model inevitably varies. Although several homogenization strategies are available (e.g., Weatherill et al., 2016), the most appropriate approach should be determined on a case-by-case basis and only after a critical analysis of the available data, and thus cannot be explicitly defined in advance. In this study, a top-down approach was taken to homogenize each data set, i.e., an initial structure was defined based on the



most uniform and non-conflicting information available across the region, and progressively more granular and detailed information was introduced, selected and ranked based on its reliability and importance to the hazard assessment. Multiple representations or interpretations of a single element that cannot be resolved by analysis were retained, as in the case of slip rate information for finite fault modelling. This epistemic variability is then passed on in the hazard calculation to represent uncertainty (using a logic tree or parametric distributions in OpenQuake).

The assembled earthquake catalog represents an important step toward a holistic analysis of the seismic characteristics of the

region. The conversion in the Mw scale greatly simplifies the integration of future data, the compilation of which can be uniformly based on the procedures presented and discussed in detail here. The major limitation of the derived earthquake catalog is probably its completeness level, the reduction of which is a future priority. However, this can only be done by integrating new data, i.e., by strengthening existing networks, which will also help to refine the selection of appropriate ground motion models for the region and encourage the development of new locally calibrated models. In addition, a revision of

historical data, which may currently be subject to large uncertainties, should be endorsed.

As for the active fault dataset, we started from existing regional compilations (GEM and AFEAD) that were already consistent for the whole area. Nevertheless, it should be noted that targeted studies on individual segments, either from the literature or from local scientific partners of the consortium, were noted with great interest but were not directly included in the initial source model at this time, mainly because of the scientific debate currently going on for some of these lineaments, or because

of the lack of complete information, or the degree of uncertainty involved. Nonetheless, we consider the current selection to be robust enough to represent the major fault systems capable of producing large destructive earthquakes, and thus a good starting point for later modifications by integrating local studies at higher resolution.

A well-known problem with this type of study is the long-term sustainability of the data. In this work, we adopted the strategy of making all data and the tools used to generate them freely available on platforms (see Data Availability section) that ensure

their continuous accessibility. Our goal is to create dynamic input datasets (i.e., both the earthquake catalog and the active fault database) and hazard models that can be easily maintained and later expanded as new information becomes available. Ultimately, the goal is to provide the local scientific community with an aggregator that can be used to foster discussion and subsequently enriched with the results of targeted studies on selected elements that require specific attention.

**Data availability**

All data presented in this paper, including the harmonized earthquake catalog, the active fault database, the PSHA source model files in OpenQuake format and the corresponding calculation results, are available on the World Bank data portal (https://datacatalog.worldbank.org) along with the technical reports produced during the SFRARR project.



## Acknowledgments

The "Strengthening Financial Resilience and Accelerating Risk Reduction in Central Asia" (SFRARR) Program is funded by
the European Union, managed by the Global Facility for Disaster Reduction and Recovery (GFDRR) and implemented by the
World Bank.

We would like to thank all the project team members, the local partners of the consortium and the World Bank specialists, in
particular Dr. Stuart Alexander Fraser and Dr. Madina Nizamitdin, for their constructive contribution to the project.

## Disclaimer

The authors of this document are part of an international consortium of experts that has been hired by the World Bank to
implement many of the work under the SFRARR Central Asia program. This model has been produced with the assistance of
the European Union, the World Bank, and GFDRR. The sole responsibility of this publication lies with the author and can in
no way be taken to reflect the views of three institutions. The European Union, the World Bank, or GFDRR are not responsible
for any use that may be made of the information contained therein.

## 440  Author contribution

VP was responsible for coordinating the earthquake hazard component of the SFRARR program. All co-authors contributed
to data collection and review, the model implementation and to the discussion of key results. VP prepared the manuscript with
contributions from all co-authors.

## Competing interests

The authors declare that they have no conflict of interest.

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
