# Peer review of "Harmonizing seismicity information in Central Asian countries: earthquake catalog and active faults"

_Natural Hazards and Earth System Sciences, 2023_

## Referee Comment (RC1)

Review of the manuscript of "Harmonizing seismicity information in Central Asian countries: earthquake catalog and active faults" by Poggi et al.

Development of a regionally consistent seismic catalog is a promising step towards advancing seismic hazard assessment and risk mitigation, which this study targets by focusing on Central Asia. In particular, open source development, and the emphasis given on a region that is diverse in terms of economical development and likely to be prone to future anthropogenic hazards due to industrial activities of the energy sector, I find the study of high value and would recommend its publication after the below comments are addressed.

Best regards,
Elif Oral

- In general:

  - An argumentation on what makes this manuscript worth publishing with respect to Poggi et al. (2023) https://doi.org/10.5194/nhess-2023-132 lacks.

  - I could not find any discussion on other groups' studies on the same region (see for example, Caravan of GFZ https://www.gfz-potsdam.de/en/section/seismic-hazard-and-risk-dynamics/data-products-services/caravan). In other words, a brief but essential discussion on how the developments in this work can be useful/transformative for different topics under seismic hazard assessment (early warning; physics-based and/or probabilistic hazard assessment, etc.) lacks.

- To give an idea about spatiotemporal variation of seismicity, using the information in Fig. 5 on a map view like Fig. 4 would be helpful.

- L280: it reads like blasting and mining explosions are not human-induced. But they are. Please verify the validity of the terms.

- L280: if only removing anthropogenic events result in Poisson process, it contradicts with the 2nd sentence of the section.

- Please verify the use of "artificial" events. Induced and triggered events are the common terms, and they both can relate to anthropogenic activities such as blasting, geothermal activities, etc. At this point, if necessary, you can distinguish induced and triggered events in parallel with literature (see McGarr et al., 2002 https://doi.org/10.1016/S0074-6142(02)80243-1).

- L305: "… the largest events in the cluster are likely to be of natural origin." A better way to justify this point is to show and comment on seismicity variation through magnitude-frequency distribution. Please quantify the variability of results to judge its significance/insignificance.

- L320: Any references of PSHA, and also for uncertainty related to source and ground motion?

- Table 9: USD: Do you mean surface or ground level? Surface rupture can relate to a given event, and does not necessarily mean surface level.

- L380: Please provide references for fault reliability and related classes.

Minor comments:

- Abstract: harmonized between countries? Do you mean countries in the same region make use of the same catalog? Unified? Regional joint use?

- Abstract: homogenized in Mw?

- L75: homogeneous Mw?

- L225: native estimate of Mw?

- I see what you mean by homogenization, but giving its definition at least once in the beginning would make it easier to follow the text.

- L395: a top-down approach?

- Line numbers for each "line" would be helpful for review.

---

## Author Comment (AC2)

**Reply to RC2**

We extend our gratitude to the anonymous reviewer for her/his positive evaluation of our work. We have carefully considered all of the suggestions provided and have addressed them comprehensively in this response. Furthermore, we have also incorporated these suggestions into the updated version of the manuscript.

1) the term homogenized instead of harmonized or explained what it means "harmonized" in "earthquake catalog harmonized between countries".

   *We use the terms "homogenisation" and "harmonisation" in slightly nuanced contexts and with different meanings, although they are closely related and often mistakenly thought to be synonymous. While homogenisation usually refers to the "process" that leads to a uniform representation, harmonisation emphasises the properties of the final product, especially in relation to its applications.*
   *In this study, for example, we have compiled a catalogue that is homogeneous in magnitude, reflecting the process of standardising scales to achieve uniformity. However, the resulting catalogue represents a harmonised dataset across countries, specifically tailored for seismic hazard analysis.*
   *Therefore, we do not use these terms interchangeably, but select them based on their contextual appropriateness. Nonetheless, we acknowledge the reviewer's concerns and have amended the manuscript accordingly to provide additional clarification to eliminate any confusion.*

2) At L 80 there is a statement - "including the description of intensity in moment magnitudes (Mw) ". I do not know formulas of Mw which include the seismic intensity, I, maybe it is better to give an example of a formula or exclude this statement if there is not the case.

   *We admit that the sentence was perhaps misleading. With the term "intensity" we wanted to refer to the "size" of earthquakes and not to the more specific "macroseismic intensity". To clarify this, we have amended the sentence as follows: "including a description of earthquake size on the moment magnitude (Mw) scale" This change aims to provide a clearer understanding of the terminology used.*

3) L350, L380 - reference to an annex to the reliability classes of faults.

   *The manuscript does lack this information, and we appreciate both reviewers bringing it to our attention. The reference to the reliability classes can be found in the documentation of the AFEAD database, specifically in section 3.3 "Characteristics of evaluated attributes," as outlined in Bachmanov et al.'s 2017 publication. However, while the original source is available solely in Russian, an English version can be accessed via the following link:*

   http://neotec.ginras.ru/index/english/database_06_eng.html

   *We have now incorporated this link into the revised version of the manuscript for readers' convenience.*

4) L390 instead of "The homogenization of input datasets" , maybe "The harmonization of input datasets". The terms seem synonyms.

   *In this case, our focus is on the complicated and time-consuming process of homogenisation and not on the resulting product. We have therefore decided to consistently use the term "homogenisation" instead of "harmonisation" in this context, in line with what was discussed in point 1 of this reply. We are confident that the clarification in the manuscript effectively recognises this distinction and helps to avoid potential misunderstandings.*

5) L395 instead of a "top-down approach" maybe a "general-particular (detailed) approach".

   *We welcome the suggestion. We have changed the definition "top-down approach" to "general-to-specific approach", which appears more appropriate.*

---

## Author Comment (AC3)

**Reply to RC1**

We would like to thank Dr. Oral for her positive evaluation of our work, whose findings will certainly contribute to improving the quality of our manuscript. We have thoroughly considered each suggestion and have taken steps to implement it wisely. In this response, we offer detailed explanations and revisions that reflect our efforts to address the reviewer's comments. In addition, we have carefully incorporated these improvements into the revised version of the manuscript to ensure that the final document reflects our commitment to excellence and our responsiveness to the feedback.

1) An argumentation on what makes this manuscript worth publishing with respect to Poggi et al. (2023) https://doi.org/10.5194/nhess-2023-132 lacks.

   *We acknowledge the observation. The Introduction has been revised to clarify the relationship between the two companion papers, and their distinct focuses. Line 67 has been amended as follows:*

   *"... to complement the observed seismicity for the construction of a comprehensive probabilistic seismic hazard model for Central Asia. Further details on this model can be found in the companion article of this Special Issue (Poggi et al. 2023), where the construction of a hybrid source (distributed seismicity and finite faults) and ground motion model is extensively described, along with a thorough discussion of the results."*

2) I could not find any discussion on other groups' studies on the same region (see for example, Caravan of GFZ https://www.gfz-potsdam.de/en/section/seismic-hazard-and-risk-dynamics/data-products-services/caravan). In other words, a brief but essential discussion on how the developments in this work can be useful/transformative for different topics under seismic hazard assessment (early warning; physics-based and/or probabilistic hazard assessment, etc.) lacks.

   *The reviewer is correct. This manuscript does indeed come from a larger project in which many local studies and regional initiatives were thoroughly considered. Unfortunately, due to space constraints, we had to exclude some components in this article, which are presented more comprehensively in the accompanying paper on probabilistic seismic hazard. As noted by the reviewer in the previous comment, we have now better described the link between the two submitted companion papers, which are indeed complementary. In addition, we have now added some relevant references for the region to the introduction as follows:*

   *"This study builds on and updates the results of previous important regional studies such as the Global Seismic Hazard Assessment Programme (GSHAP, Giardini et al., 1999; Ulomov et al., 1999) and the EMCA project ("Earthquake Model of Central Asia"," see Bindi et al., 2011, 2012; Ullah et al., 2015). It also considers recent progress at the national level, including the work of Ischuk et al. (2014, 2018) for Kyrgyzstan, Tajikistan and Eastern Uzbekistan, Silacheva et al. (2018) and Mosca et al. (2019) for Kazakhstan, the Central Asia Seismic Risk Initiative (CASRI, Abdrakhmatov, 2009) for Kyrgyzstan and Artikov et al. (2018) for Uzbekistan, among others."*

3) To give an idea about spatiotemporal variation of seismicity, using the information in Fig. 5 on a map view like Fig. 4 would be helpful.

*We have indeed explored this possibility, as we recognize the potential utility of displaying spatial and temporal variability together on a single plot. To this end, we have created a figure (see below). However, upon examination, the results do not appear to be as informative as we initially anticipated. In fact, we are concerned that the figure may unintentionally give the misleading impression that only older events cluster around the main active structures. This effect is evidently influenced by the completeness of the catalogue and the order of overlapping visualisation layers. Consequently, we are not entirely certain whether it would be advisable to include this new figure in the manuscript. Would it be perhaps feasible to include it as an addendum, e.g. in the form of electronic supplement?*

[Figure]

4) L280: it reads like blasting and mining explosions are not human-induced. But they are. Please verify the validity of the terms.

*We understand the reviewer's concern; however, we are unable to pinpoint where in the manuscript this potential misinterpretation could occur. In essence, we have classified events into three main categories: i) purely natural events, ii) events originating from natural sources but triggered by human activity, and iii) events stemming from purely artificial sources. Mining and explosions fall into the latter category, which is unequivocally human induced.*
*We tentatively propose the following sentence modification to enhance clarity:*

*"The clustering of correlated events may be of natural origin (e.g., the aftershocks following a major event), be caused by the interference between human activity and the natural*

*environment (e.g., geothermal exploitation - extraction of thermal energy by pumping fluids from a geothermal reservoir and carbon sequestration - process of capturing and storing atmospheric carbon dioxide in an already depleted reservoir), or purely anthropogenic, such as those originated by the use of artificial sources (e.g., blasting, mining explosions)."*

5) L280: if only removing anthropogenic events result in Poisson process, it contradicts with the 2nd sentence of the section.

*The sentence refers to all correlated events that must be eliminated to achieve (ideally) Poissonianity, and not solely those of anthropogenic origin. To improve clarity and avoid possible misunderstandings, we have revised the sentence as follows:*

*"All correlated events (both natural and associated with human activity) must be removed..."*

6) Please verify the use of "artificial" events. Induced and triggered events are the common terms, and they both can relate to anthropogenic activities such as blasting, geothermal activities, etc. At this point, if necessary, you can distinguish induced and triggered events in parallel with literature (see McGarr et al., 2002 https://doi.org/10.1016/S0074-6142(02)80243-1).

*In our opinion, although both referred to anthropogenic activity, one must distinguish between merely artificial events, where "artificial" is used to indicate the use of artificial sources (e.g., explosions), and induced/triggered events, which are related to the human interaction with the natural environment (e.g., geothermal fields, depleted reservoirs). In this latter case, although triggered by human activity, the sources are (partially or entirely) natural.*
*As correctly pointed out by the reviewer and in McCarr et all. 2002, a further distinction should be made between induced and triggered, although both falling in the above definition. For the application of this regional study, however, this level of discrimination does not appear necessary or feasible, as it would require further resources, which are presently not available.*
*It must be noted, moreover, that in Central Asia countries, most of the anthropogenic events in catalogue are of the former type (purely artificial), given the well-known ongoing mining activities in the region. Industrial facilities that could trigger induced seismicity are virtually (at least to our knowledge) inexistent.*

7) L305: "... the largest events in the cluster are likely to be of natural origin." A better way to justify this point is to show and comment on seismicity variation through magnitude-frequency distribution. Please quantify the variability of results to judge its significance/insignificance.

We appreciate the reviewer's suggestion. However, as illustrated in Figure 4 of the companion paper to this article, the Magnitude Frequency Distribution (MFD) of Group E (stable continental) presents challenges in calibration due to the limited number of

available events, making quantitative comparison difficult. The MFD of zone 57, where most clusters are located, is also presented below for reviewer's convenience. After applying the declustering procedure, the number of remaining events, which are ideally considered natural, is too small to perform a reliable rate analysis. Consequently, the comparison with background seismicity had to be qualitative, primarily based on spatial distribution patterns rather than temporal occurrence.

[Figure]

8) L320: Any references of PSHA, and also for uncertainty related to source and ground motion?

*We have now incorporated the following references that support the present use of finite fault models in PSHA practice, for readers convenience. However, a more comprehensive list of references will be included in the companion paper of this special issue (presently under review), which focuses specifically on the PSHA model and its results.*

- *Danciu L, Şeşetyan K, Demircioglu M, Gülen L, Zare M, Basili R, et al. (2017) The 2014 Earthquake Model of the Middle East: seismogenic sources, Bulletin of Earthquake Engineering, Volume 16, pp. 3465–3496 (doi:10.1007/s10518-017-0096-80)*
- *Valentini, A., Pace, B., Boncio, P., Visini, F., Pagliaroli, A., Pergalani, F. (2019). Definition of Seismic Input From Fault-Based PSHA: Remarks After the 2016 Central Italy Earthquake Sequence. Tectonics, Volume 38, 2 (doi:10.1029/2018TC005086)*
- *Poggi, V., Garcia-Peláez, J., Styron, R., Pagani, M., Gee, R. (2020). A probabilistic seismic hazard model for North Africa. Bulletin of Earthquake Engineering, 18(7), pp. 2917–2951 (doi: 10.1007/s10518-020-00820-4).*
- *Gómez-Novell O, García-Mayordomo J, Ortuño M, Masana E and Chartier T (2020) Fault System-Based Probabilistic Seismic Hazard Assessment of a Moderate Seismicity Region: The Eastern Betics Shear Zone (SE Spain). Front. Earth Sci. 8:579398 (doi: 10.3389/feart.2020.579398)*

9) Table 9: USD: Do you mean surface or ground level? Surface rupture can relate to a given event, and does not necessarily mean surface level.

*Indeed, this is true. OpenQuake offers flexibility in configuring such parameter. Typically, this distinction becomes significant in site-specific models, where near-fault ground motion significantly influences the results. However, for the standard engineering application of a regional Probabilistic Seismic Hazard Analysis (PSHA) model, the simplification of using ground level appears to be adequate.*

10)     Please provide references for fault reliability and related classes.

*The manuscript does lack this information, and we appreciate both reviewers bringing it to our attention. The reference to the reliability classes can be found in the documentation of the AFEAD database, specifically in section 3.3 "Characteristics of evaluated attributes," as outlined in Bachmanov et al.'s 2017 publication. However, while the original source is available solely in Russian, an English version can be accessed via the following link:*

*http://neotec.ginras.ru/index/english/database_06_eng.html*

*We have now incorporated this link into the revised version of the manuscript for readers' convenience.*

**Minor comments:**

1) Abstract: harmonized between countries? Do you mean countries in the same region make use of the same catalog? Unified? Regional joint use?

*Please refer to our response to minor comment #5, in which we formally address the distinction between "harmonised" and "homogenised", as used in the manuscript.*

2) Abstract: homogenized in Mw?

*Please refer to our response to minor comment #5, in which we formally address the distinction between "harmonised" and "homogenised", as used in the manuscript.*

3) L75: homogeneous Mw?

*Please refer to our response to minor comment #5, in which we formally address the distinction between "harmonised" and "homogenised", as used in the manuscript.*

4) L225: native estimate of Mw?

*We acknowledge that "native estimate" is misleading. We have revised the sentence to read: "events with a direct estimate of Mw (from waveforms) are limited."*

5) I see what you mean by homogenization, but giving its definition at least once in the beginning would make it easier to follow the text.

*We use the terms "homogenisation" and "harmonisation" in slightly nuanced contexts and with different meanings, although they are closely related and often mistakenly thought to be synonymous. While homogenisation usually refers to the "process" that leads to a uniform representation, harmonisation emphasises the properties of the final product, especially in relation to its applications.*

*In this study, for example, we have compiled a catalogue that is homogeneous in magnitude, reflecting the process of standardising scales to achieve uniformity. However, the resulting catalogue represents a harmonised dataset across countries, specifically tailored for seismic hazard analysis.*

*Therefore, we do not use these terms interchangeably, but select them based on their contextual appropriateness. Nonetheless, we acknowledge the reviewer's concerns and have amended the manuscript accordingly to provide additional clarification to eliminate any confusion.*

6) L395: a top-down approach?

*A similar concern was raised by the second anonymous reviewer. We have changed the sentence to "general to specific approach", which seems more appropriate.*

7) Line numbers for each "line" would be helpful for review.

*We completely agree. Unfortunately, this is prescribed by the journal's policy, and the format of the manuscript is determined by its template. I apologize for any inconvenience this may cause.*